# The Multifunctional Protein Syntenin-1: Regulator of Exosome Biogenesis, Cellular Function, and Tumor Progression

**DOI:** 10.3390/ijms24119418

**Published:** 2023-05-29

**Authors:** Kwang-Min Lee, Eun-Chan Seo, Jeong-Hyung Lee, Hyo-Jin Kim, Cheol Hwangbo

**Affiliations:** 1Division of Life Science, College of Natural Sciences, Gyeongsang National University, Jinju 52828, Republic of Korea; stare4567@naver.com (K.-M.L.); eunchan3927@gmail.com (E.-C.S.); 2Division of Applied Life Science (BK21 Four), Research Institute of Life Sciences, Gyeongsang National University, Jinju 52828, Republic of Korea; 3Department of Biochemistry (BK21 Four), College of Natural Sciences, Kangwon National University, Chuncheon 24414, Republic of Korea; jhlee36@kangwon.ac.kr

**Keywords:** syntenin-1, exosome, microRNA, tumor microenvironment, trafficking

## Abstract

Syntenin acts as an adaptor and scaffold protein through its two PSD-95, Dlg, and ZO-1 (PDZ) domains, participating in multiple signaling pathways and modulating cellular physiology. It has been identified as an oncogene, promoting cancer development, metastasis, and angiogenesis in various carcinomas. Syntenin-1 is also associated with the production and release of exosomes, small extracellular vesicles that play a significant role in intercellular communication by containing bioactive molecules such as proteins, lipids, and nucleic acids. The trafficking of exosomes involves a complex interplay of various regulatory proteins, including syntenin-1, which interacts with its binding partners, syndecan and activated leukocyte cell adhesion molecule (ALIX). Exosomal transfer of microRNAs, a key cargo, can regulate the expression of various cancer-related genes, including syntenin-1. Targeting the mechanism involving the regulation of exosomes by syntenin-1 and microRNAs may provide a novel treatment strategy for cancer. This review highlights the current understanding of syntenin-1’s role in regulating exosome trafficking and its associated cellular signaling pathways.

## 1. Introduction

Intercellular communication is a physiological function for cell-to-cell signal sharing caused by ligand-receptor interaction, junctions, synapses, extracellular vesicles, etc. [1]. Extracellular vesicles (EVs) surrounded by lipid layers are known to transport various cargoes such as lipids, nucleic acids, and proteins to local and distant cells. EVs are classified into four categories according to size and biogenesis, or release pathways [2]. The largest EVs, the oncosomes, are 1–10 μm in size, are released from cancer cells, and generally transport large molecules such as oncogenes [3,4]. The apoptotic bodies are specifically released from dying cells and refer to an EV measuring 50 nm–2 μm [5]. Microvesicles are 100 nm–1 μm sized EVs that carry cytoplasmic molecules and are shed directly from the cell membrane [6]. Exosomes are the smallest EVs, with a size of 40–120 nm, and are produced and released by the fusion of multivesicular bodies and cell membranes [7]. Exosomes are generated in the form of intraluminal vesicles (ILVs) within the multivesicular body (MVB) of the late endosome and released into the extracellular space [8,9]. Among the endosomal sorting complex required for transport (ESCRT) that acts on the formation of ILVs, ESCRT-0, -I, and -II proteins promote the formation and bud of vesicles in MVB, and ESCRT-III induces secretion into extracellular space by cutting off buds and separating them from MVB [10]. Exosomes are transferred to recipient cells and affect target cells through binding to receptors or internalization [11]. As a result, exosomes play an important role in signaling to surrounding or distant cells, and studies of exosome formation and secretion regulators are significant for understanding intercellular communication. Syntenin is a protein that has been identified as a key player in exosome biogenesis, which is the process by which exosomes are formed and released from cells [12].

Syntenin-1 has been shown to interact with several binding partners that are involved in exosome biogenesis [13,14,15]. The combination of partner proteins activated leukocyte cell adhesion molecule (ALIX) and ESCRT-III is involved in the sorting of exosome cargo and the secretion of exosomes, respectively [16]. Syntenin-1 has also been shown to interact with syndecans, which are transmembrane proteoglycans that regulate the secretion of exosomes from cells [17]. In addition to its binding partners, syntenin-1 has been found to regulate various signaling pathways that play important roles in exosome biogenesis [12]. For instance, syntenin-1 regulates the FAK-Src signaling pathway, which is important for cell adhesion and growth regulation, through interaction with integrin [18,19]. Moreover, syntenin-1 regulates the mTOR signaling pathway, which is involved in the regulation of cell proliferation and may also play a role in exosome biogenesis [20,21]. Additionally, the Wnt/β-catenin pathway, which plays a crucial role in cell differentiation and proliferation, has been reported to be regulated by syntenin-1 and may also contribute to exosome biogenesis and release [22].

Furthermore, recent studies have suggested that syntenin-1 can transport specific miRNAs, which are small non-coding RNAs that regulate gene expression [23]. Specific miRNAs have been shown to be selectively packaged into exosomes, and syntenin-1 may play a role in the sorting and transport of these miRNAs into exosomes [24]. This suggests that syntenin-1 may have an important role in regulating intercellular communication through the transfer of specific miRNAs via exosomes.

Exosomes are an important factor in intercellular communication and have recently been attracting attention due to their influence on cancer cells and surrounding cells in the cancer microenvironment. Syntenin acts as a major regulator of exosome synthesis, and it is overexpressed in various tumors. Therefore, understanding the role of syntenin-1 in exosome biogenesis and its interactions with binding partners, signaling pathways, and miRNAs may provide important insights into the mechanisms of intercellular communication, and could lead to the development of novel therapeutic strategies targeting this process.

## 2. Syntenin Structure

Syntenin was first identified as a binding partner of the C-terminal cytoplasmic domain of syndecan, so it was also called syndecan binding protein (SDCBP). In addition, it is also called melanoma differentiation-associated gene-9 (MDA-9) because it has been confirmed that syntenin expression is regulated by the treatment of mezerein and interferon-gamma in human melanoma cells [25]. Human syntenin is composed of 298 amino acid residues, and the sequence is well preserved across all species. Syntenin is composed of four domains: the N-terminal domain (NTD), the first PSD-95, Dlg, and ZO-1 (PDZ) domain (PDZ1), the second PDZ domain (PDZ2), and the C-terminal domain (CTD) (Figure 1). Typically, the PDZ domain is located in the intracellular or plasma membrane and binds to the C-terminal peptide of the partner protein.

Although few studies have focused on the NTD and CTD regions, the NTD, consisting of 100 amino acids, holds autoinhibitory sites, recruits Sox4, and interacts with ubiquitin [26]. PDZ1 and PDZ2 have differences in binding partners and recognize and bind the c-terminal sequence motif of the target protein. In the case of PDZ1, the binding site is surrounded by Lys124, Arg128, Lys130, His175, and Lys179 to exhibit positive charges, whereas in the case of PDZ2, it does not have positive charges [27]. Because PDZ1 has weak binding affinity to target proteins, PDZ2 is the preferred binding site for most partner proteins. PDZ1i, which binds to the PDZ1 domain of syntenin to prevent binding to other proteins, acts as an antagonist of syntenin [28]. In prostate cancer, glioblastoma, and neuroblastoma, cancer metastasis is reduced by PDZ1i, which is caused by inhibition of insulin-like growth factor-1 (IGF-1), FAK, and integrin signaling, respectively [29,30,31]. Although research on syntenin inhibitors has not yet been actively conducted, further research is needed as syntenin is a major protein in cancer metastasis.

## 3. Binding Partners

Syntenin-1 is a protein that interacts with various proteins to regulate cellular functions, including exosome production, cell motility, tumor metastasis, neurotransmitter transport, and immune response (Figure 2). The following proteins are known to interact with syntenin-1.

### 3.1. ALIX and c-Src

Exosome production by syntenin-1 depends on its PDZ domain interaction with ALIX, a component of the ESCRT that facilitates membrane budding and scission for exosome generation. The three LYPX(n)L motifs in syntenin-1’s PDZ domain directly bind with ALIX to regulate exosome biogenesis [16,32,33,34]. In addition, recent studies have identified endoglycosidase heparanase as a stimulator for the syndecan-syntenin-1-ALIX pathway. Endoglycosidase heparanase increases endosomal syndecan levels by promoting internalization, leading to syndecan clustering and the increased formation of syntenin-1-ALIX-ESCRT complexes [35].

The exosome formation by the syntenin-ALIX complex is promoted by phosphorylation, and c-Src is known as the controlling kinase of this process. c-Src is a proto-oncogene tyrosine-protein kinase that binds to syntenin-1, and the reduction in c-Src inhibits exosome synthesis [14]. c-Src phosphorylates Y46 of syntenin-1, and the phosphomimetic form of syntenin-1 increases the production of exosomes. c-Src-mediated phosphorylation indicates the need for exosome synthesis because endosomal budding is retained when both syndecan and syntenin-1 are phosphomimetic forms, while it is reduced by c-Src inhibition. The study by Hikita et al. has shown that c-Src binds to ALIX to promote exosome synthesis, confirming that c-Src is a crucial regulator for exosome synthesis through syndecan-syntenin-1-ALIX signaling [36]. The interaction of syntenin, ALIX, and c-Src is a major step in regulating the formation of exosomes in cells.

### 3.2. GTPase

GTPases are hydrolase enzymes that bind to the nucleotide guanosine triphosphate (GTP) and hydrolyze it to guanosine diphosphate (GDP), and they act as regulators of various cellular processes [37,38]. Syntenin-1 induces the proteasomal degradation of the Ras homolog enriched in the brain (Rheb) through interaction, which promotes spin synapse formation [39]. Another GTPase, ADP Ribosylation Factor 6 (Arf6), acts as an important regulator at the stage of releasing vesicles formed within cells into the extracellular space. Arf6 can form complexes with syntenin-1 through phosphatidylinositol 4,5-bisphosphate (PIP2), and this complex acts as an essential factor in epiboly and the early development of zebrafish [40]. In addition, the activity of Arf6 by syntenin-1 and the PIP2 complex regulates cell membrane dynamics [41]. Arf6 and its effector phospholipase D2 (PLD2) act as regulators of exosome biogenesis by controlling the budding of intraluminal vesicles (ILVs) through the syntenin-1-ALIX pathway [15].

### 3.3. Merlin

Merlin, also known as neurofibromatosis 2 (NF2) or schwannomin, is a cytosolic tumor suppressor that can cause NF type II when mutated [42]. Its structure is similar to that of Ezin, Radixin, and Moesin (ERM), which is a member of the protein 4.1 superfamily (erythrocyte membrane skeleton) [43]. The ERM protein acts as a molecular adaptor that connects integral membrane proteins to the cytoskeleton and is involved in regulating cell shape, membrane specialization, and the distribution of adhesion molecules to the cell surface [44,45]. Syntenin-1 binds to merlin’s c-terminal, and when this interaction occurs, merlin is observed in the plasma membrane and cytoplasm of the filopodia’s extension region in Hela cells. Conversely, when syntenin-1 levels decrease, cells show a flat shape, and merlin decreases in the plasma membrane [46]. The syntenin-1-mediated changes in the plasma membrane indicate that cell motility can be regulated in tumor metastasis. Although it has been confirmed through protein analysis that merlin is released through exosomes from brain cancer cells, the exact release mechanism has not yet been studied [47].

### 3.4. Glycine Transpoter Subtype 2 (GlyT2) and Syntaxin

Glycine is a major neurotransmitter in inhibitory synapses of the brainstem, and its transport is facilitated by glycine transporters present in the plasma membrane [48]. The C-terminal region of GlyT2 contains TQCs, which are required for binding to syntenin-1 [49]. Both PDZ domains of syntenin-1 are essential for the transmission of signals from GlyT2, and the binding force of each domain is different. PDZ2 is crucial for the binding of syntenin-1 to GlyT2C, and PDZ1 and PDZ2 must be present together for a complete interaction to occur [49]. When GlyT2 is in synaptic vesicles and syntaxin is present, the vesicles are trafficked to the plasma membrane [50]. However, since syntenin-1 cannot bind to syntaxin and GlyT2 simultaneously, it is believed that syntaxin reduces the interaction between syntenin-1 and GlyT2, resulting in vesicle internalization of GlyT2.

### 3.5. CD63

Tetraspanin, a protein that consists of various families involved in various functions such as immune responses, tumor cell migration, and exosome abundance, binds directly to syntenin-1 through the cytoplasmic tail of the C-terminal [51,52]. CD63 belonging to Tetraspanin binds directly through the cytoplasmic tail of the C-terminal when binding with syntenin-1 [53]. The interaction is possible for both PDZ1 and PDZ2 of syntenin-1, but chemical shift perturbations occur mainly in Arg119, Asp120, Val141, Ala143, Arg176, Leu127, Arg128, and Arg179 of PDZ1. In addition, a C-terminus mutation of syntenin-1 reduces binding stability with CD63 [53]. CD63 also plays a role in disease by binding to the capsid protein of human papillomaviruses (HPV) and trafficking to multivesicular endosomes, but this function is impaired when the combination of CD63 and syntenin-1 is reduced [54].

### 3.6. Interleukin-5 Receptor a (IL5Ra)

IL5Ra functions as a regulator of myeloid cell differentiation and leukocyte function [55]. Syntenin-1 interacts with the C-terminus of IL5Ra through its PDZ domain, which can bind to IL5Ra with similar affinity [26]. The combination of the two PDZ domains of syntenin-1 facilitates the formation of the IL5Ra signaling complex [56]. SOX4, a transcription factor activated by IL5, is protected from proteasomes and increases in expression due to its direct binding to syntenin-1, which regulates its activity [26,57]. Computational analysis revealed that 31 genes, including IL-13 and IL-5Ra, are putative targets of the miRNAs up-regulated in airway-secreted EVs (AEVs) but down-regulated in lung tissues after house dust mite (HDM) exposure [58].

## 4. Cellular Signaling Pathways

Syntenin-1 has been implicated in various cellular signaling pathways associated with exosome biogenesis, trafficking, cell motility, and adhesion (Figure 3). In the Wnt signaling pathway, syntenin-1 interacts with Wnt receptors and facilitates their internalization into endosomes. Syntenin-1 serves as an adaptor protein for integrin β1 in integrin signaling, activating integrin linked kinase (ILK)-mediated signaling and modulating the formation of integrins β1-FAK-Src, thereby enhancing cell mobility and invasion. In CD98 signaling, CD98hc binds to integrins β1 and β3 to activate signaling that promotes cell proliferation and migration. The binding of syntenin-1 and PKCα is critical for the activation of FAK by integrin signaling. Conversely, CD99 disrupts the PKCα-syntenin-1-FAK complex, thereby reducing the formation of focal adhesions. Syntenin-1 also interacts with transforming growth factor (TGF)-βI; in TGF-β signaling, stabilizing signaling and promoting metastasis and invasion in cancer cells. Syntenin-1 activates Smad by TGF-β1, leading to epithelial–mesenchymal transition (EMT), invasion, and metastasis.

### 4.1. Wnt Signaling

Wnt signaling is involved in cell proliferation, motility, and embryonic development, and its activation is based on the binding of Wnt proteins to Frizzled receptors [59,60]. This signaling cascade is divided into a canonical pathway (β-catenin-dependent) and a non-canonical pathway (β-catenin-independent) through Dishevelled (Dsh) activation [61]. The canonical pathway of Wnt activates β-catenin-dependent transcription and regulates cell differentiation, growth, and survival [60]. The non-canonical pathway controls cytoskeleton reorganization, migration, and polarity via GTPase and PKC activation [62,63]. 

The binding of Frizzled 7 and syntenin induces PKCα/Cdc42 noncanonical Wnt signaling cascades and extension movements in Xenopus [22]. In human cells, syntenin-1 interacts with c-terminal Frizzled 7 and PIP2. The complex of syntenin-1-PIP2-Frizzled 7 mediates Frizzled 7 trafficking to the plasma membrane through an ARF6/PIP2-dependent pathway [64]. These studies suggest that syntenin-1 selects intracellular molecules as cargo through binding and plays a pivotal role in trafficking.

### 4.2. Integrin Signaling

Integrin is a cell-surface receptor that mediates cell-extracellular matrix proteins consisting of α and β subunits [65]. The receptor exists in a bent form with low binding force to the ligand, but inside-out signaling exposes the binding site to the ligand. Talin, belonging to inside-out signaling, promotes binding to ECM molecules by exposing the binding site of the integrin receptor [66]. This signaling induces binding to cytoplasmic focal-adhesion proteins, including ILK, PINCH, and Parvin, to form the IPP complex [67]. ILK is a major member of the IPP complex and activates signaling that induces gene expression related to cell adhesion and growth.

Syntenin-1 acts as an adaptor protein for integrin β1 and regulates the formation of the IPP complex and activation of ILK-mediated signaling. It acts as a scaffold protein to mediate the activation of AKT through phosphorylation by ILK and induce activation of integrin β1 and the downstream factor FAK [18,68]. Expression of syntenin-1 by the proteinoglycan co-receptor syndecan 4 is essential in this process [69]. Syntenin-1-fibronectin binding modulates the formation of integrin β1-FAK-Src, thereby enabling signaling to improve cell mobility and invasion capabilities [19].

### 4.3. CD98, 99-Mediated Signaling

#### 4.3.1. CD98

CD98 is a 125-kDa heterodimeric type II transmembrane glycoprotein that consists of an 85-kDa glycosylated heavy chain (CD98hc) and a non-glycosylated 40-kDa light chain, which plays a crucial role in amino acid transport and integrin signaling [70]. CD98hc activates signaling pathways involved in cell proliferation and migration by binding to integrins β1 and β3 [71]. The activation of FAK by integrin signaling requires the critical interaction between syntenin-1 and PKCα. CD98 promotes the phosphorylation of Y397 in FAK, which enhances the interaction of PKCα-syntenin-1-FAK, focal adhesion, and the formation of the IPP-AKT-syntenin-1 complex [19,72]. The phosphorylation of FAK by CD98 forms SH2-mediated binding sites that bind Src-family tyrosine kinases, leading to the formation of the FAK-Src-integrin β1 complex [72,73]. Syntenin-1 facilitates the interaction between ILK and AKT, and in this process, CD98-induced phosphorylation of PKCα promotes the complex formation of integrin and IPP, thereby transmitting sub-signals [18]. 

#### 4.3.2. CD99

CD99, a transmembrane glycoprotein highly expressed in various cell types such as peripheral T cells and hematopoietic cells, has been shown to play a crucial role in lymphocyte differentiation, activation, and proliferation [74,75,76]. It acts as a modulator of cellular processes by recruiting SHP2, a phosphatase, to the plasma membrane, leading to its activation. SHP2 subsequently dephosphorylates FAK, disrupts the PKCα-syntenin-1-FAK complex, and reduces the formation of focal adhesions [77,78,79]. By interfering with the formation of syntenin-1-activated complexes, CD99 exerts a negative regulatory effect on cell adhesion and migration. 

### 4.4. TGF-β Signaling

Mammalian TGF-β comprises three isoforms (TGF-β1, 2, and 3) and serves as a multifunctional cytokine that participates in various physiological processes, including cell growth, differentiation, death, and migration [80]. TGF-β signals through its binding to TGF-βR, which has two isoforms, TGF-βRI; and TGF-βRII, and contains a terminal, transmembrane, and kinase domain [81]. 

The canonical pathway of TGF-β signaling commences with the phosphorylation of SMAD2 and SMAD3 by TGF-βR, which is activated by the binding of TGF-β ligand. SMAD4 is subsequently recruited and translocated into the nucleus. On the other hand, SMAD7 binds to activated TGF-βR or phosphorylated SMAD2/3 and induces degradation via SMAD-specific E3 ligase. In this process, syntenin-1 interacts with TGF-βI; to stabilize signaling and promote metastasis and invasion in cancer cells through activation of the GTPases RhoA and Cdc42 [82]. The combination of syntenin-1 and TGF-β1 disrupts the interaction between caveolin-1 and TGF-βRI, leading to an increase in SMAD2/3 activation and inhibition of caveolin-1 internalization. TGF-β signaling is inhibited by the internalization of TGF-β1 using flotillin-rich lipid rafts and caveolin-rich caveolae. However, the combination of syntenin-1 and TGF-β1 inhibits the interaction with caveolin-1 and TGF-βRI, disrupting the internalization via caveolin and increasing the phosphorylation and activation of SMAD2. The activation of TGF-β signaling by syntenin-1 increases EMT in cancer cells [83]. Hence, syntenin-1 activates SMAD by TGF-β1, resulting in EMT, invasion, and metastasis [84]. 

## 5. Function of Syntenin

Syntenin-1 is a protein that plays important roles in various cellular processes such as protein trafficking, cell motility, and exosome formation. Syntenin-1 also interacts with delta1 and regulates its localization on the cell surface, which is important for the activation of the Notch pathway. In cell motility, syntenin-1 interacts with c-Src and phosphorylated-FAK (p-FAK) to promote cell migration through the activation of the NF-kB pathway and MMP2 gene expression. Syntenin-1 is also involved in the organization of adherens and the regulation of membrane dynamics. In exosome formation, syntenin-1 interacts with syndecan and ALIX to promote the formation of ILVs and the release of exosomes. Syntenin-1 plays a crucial role in regulating intercellular communication and cellular functions through these various cellular processes.

### 5.1. Protein Trafficking

Syntenin-1 has been reported to bind with syndecan, a heparan sulfate proteoglycan that regulates various growth factors and cell adhesion molecules and is involved in the endosomal recycling process [85]. Ptdlns (4)P 5-kinase activity and the rotation rate of Ptdlns (4,5)P2 are crucial for endosome trafficking to the plasma membrane in the endosomal recycling pathway. The accumulation of syntenin-1 and syndecan in the endosome is induced by an increase in Ptdlns(4)P 5-kinase and Ptdlns(4,5)P2, leading to their binding to the perinuclear recycling component. This signaling pathway is essential for the recycling of syndecan, and the increase in Ptdlns(4,5)P2 is critical for the accumulation process [41].

Notch receptor, upon binding with ligands such as Delta1, undergoes cleavage, leading to the translocation of the intracellular domain of the notch protein (NICD) to the nucleus, where it acts as a transcription activator, regulating the cell differentiation process [86,87]. Delta1 has a PDZ-binding motif in its C-terminal and has been identified as a binding partner of syntenin-1 through two-hybrid screening [88]. Syntenin-1 plays a crucial role in intracellular protein trafficking, including membrane targeting and recycling, by promoting the accumulation of Delta1 at the plasma membrane. Knockdown of syntenin-1 reduces Delta1 on the cell surface, indicating its crucial role in protein trafficking.

### 5.2. Cell Motility and Membrane Organization

c-Src is a kinase known to promote cell motility, invasiveness, and malignant progression in diverse types of cancer cells [89]. The induction of cell migration by c-Src is facilitated by its binding with phosphorylated focal adhesion kinase (p-FAK), and this migratory capacity is reduced by knockdown of syntenin-1 or overexpression of dominant-negative FAK [68]. c-Src and FAK signaling activate the NF-kB pathway involved in the transcriptional activity of invasion-related genes [90]. NF-kB acts as a transcription activator and regulates the expression of the MMP2 gene involved in cell migration and invasion [68]. MMP2 has been shown to increase invasion and metastasis in various types of cancer, including small cell lung cancer, melanoma, and hepatoma cells [91,92,93]. Because c-Src regulates cell mobility, cytoskeletal rearrangement, and membrane architecture via various signaling pathways, interaction between c-Src and syntenin-1 is likely to regulate cell movement and invasion through different pathways [73,94]. 

Syntenin-1 is known to co-localize with syndecan-1, e-cadherin, α-catenin, and β-catenin in filopodia, which are actin-rich plasma membrane protrusions of epithelial cells. Through its interaction and localization, syntenin-1 can act as a potential regulator of the organization of adherens and the regulation of membrane dynamics [95]. In addition, syntenin-1 in zegrafish plays an important role in the early development process by regulating epiboly progression through interactions with PIP2, Arf6, and syndecan [40]. These findings suggest that syntenin-1 plays a major role in cell motility and membrane formation not only in various cells but also in cancer cells.

### 5.3. Autophagy

Autophagy is a system that breaks down unnecessary substances to maintain the homeostasis of cells under stress conditions such as starvation and hypoxia [96]. Autophagy and exosome production control each other through various processes and can compensate each other in cell stress conditions [97]. Down-regulation of autophagy-related proteins 16-like 1 (ATG16L1) and autophagy-related proteins 5 (ATG5), the main regulators of autophagy, inhibits exosome synthesis and reduces metastasis in breast cancer [98]. In addition, the reduction in the ATG12-ATG3 complex involved in late autophagosome formation decreased the formation of exosomes [99]. Conversely, autophagy and exosome formation can act as antagonists through amphisomes [100]. Amphisomes formed through fusion between autophagosomes and MVBs reduce exosome release [101]. In addition, autophagy can inhibit exosome biogenesis by promoting the lysosome-dependent decomposition of MVB [102].

Syntenin-1 has been reported to regulate anokis resistance and cell death through protective autophagy regulation in glioma stem cells (GSCs) [103]. The activity of PKC, epidermal growth factor receptor (EGFR), and B-cell lymphoma 2 (BCL2) through the binding between FAK and syntenin maintains protective autophagy and prevents cell death in anoikis. Syntenin-1 can act as a factor that regulates autophagy, and autophagy is an important phenomenon that can control the production and secretion of exosomes. Further research is needed on the role of syntenin-1 in the regulation of exosomes through autophagy, which can identify the new role of syntenin-1 in the cancer microenvironment.

### 5.4. Exosome

#### 5.4.1. Exosome Synthesis

Exosomes are small extracellular vesicles, ranging from 40 to 120 nm, that are produced by all cell types and contain nucleic acid in the form of DNA/RNA, protein, and microRNA, which can regulate various cellular functions [12,104,105]. As secretory vesicles, exosomes play a critical role in intercellular communication, including neuron–neuron communication, and are involved in physiological processes such as inflammation, angiogenesis, and neurodegenerative diseases [106,107,108]. The exosomes or microvesicles secreted by tumor cells have the ability to induce an immunosuppressive microenvironment, encourage tumor cell migration, and aid in metastasis [109].

When quantitative proteomic analysis was performed on exosomes separated from various cell lines, it was confirmed that the expression of syntenin-1 was the highest [110]. Syntenin-1, along with syndecan and ALIX, has been identified as an important protein in the formation of exosomes [16,111]. Syndecan is a type-1 membrane-spanning protein with an intracellular domain (ICD) and heparan sulfate (HS) chains, which can bind to ligands such as FGF2 and fibronectin on the cell surface and regulate the intracellular environment and migration according to the ligand [112,113,114]. The interaction of syntenin-1’s PDZ domain with PIP2, regulated by Arf6 and PIPK activation, induces intracellular recycling of syndecan and the FGF2-FGFR1 complex to the cell surface via the late recycling endosome [41]. The syntenin-1-syndecan-ALIX complex promotes the formation of ILVs and facilitates the budding and release of endosomes for the sorting of specific cargo [34]. Overall, the FGF2-syndecan-syntenin-1-ALIX complex promotes the release of FGFR1 through the generation of exosomes. Through these signaling pathways, syntenin-1 serves as a regulator in the formation of syndecan-associated exosomes (Figure 4).

#### 5.4.2. Exosome Endocytosis

The internalization of exosomes occurs via recognition by heparan sulfate proteoglycans (HSPG) or receptors and is similar to the pathway of virus infection [115,116]. Syntenin-1 knockout decreases HSPG levels, leading to reduced viral transduction and exosome uptake [117,118]. Furthermore, reduced levels of syntenin-1 inhibit the virus endocytosis pathway by impeding the formation of ILVs. Syntenin-1 has been shown to bind to nectin-1, a receptor for herpes simplex virus (HSV), but further investigation is required to determine its intracellular effects [119].

## 6. Relationship between Syntenin and microRNA

MicroRNAs (miRNAs or miRs) are small, non-coding RNA molecules consisting of 21–25 nucleotides involved in various cellular signaling processes [120,121]. miRNA binds to a sequence in the 3’ untranslated region (UTR) of mRNA or non-coding RNA to induce degradation [122]. miRNA is first transcribed by RNA polymerase II as a pre-miRNA with a hairpin structure, 5’-cap, and poly(A)-tail. Then, the pre-miRNA is processed by the Drosha/DGCr8/Pasha complex and transported out of the nucleus via Exportin-5/Ran-GTP [123]. In the cytoplasm, the pre-miRNA is cleaved by Dicer (RNase III) to generate mature miRNA, which then binds to RNA through the RNA-induced silencing complex (RISC) and inhibits translation [124,125].

### 6.1. The Regulation Mechanism of Syntenin by miRNA

*SDCBP* is a coding gene for syntenin-1 and is highly expressed in various types of cancer [126]. The expression of syntenin-1 is regulated by the silencing of the *SDCBP* gene by various miRNAs (Table 1).

**Table 1 ijms-24-09418-t001:** List of miRNAs that regulate syntenin.

Micro RNA	Region	Cancer	Reference
miR-135-5p	3’UTR	breast cancer	[127]
miR-216b	3′UTR	breast cancer	[128]
miR-216b	3′UTR	pancreatic cancer	[129]
miR-1273a	3′UTR	Lung cancer	[130]
miR-361-5p	3′UTR	gastric carcinoma	[131]
miR-23a	3′UTR	Melanoma	[132]
miR-139-3p	3′UTR	Gliomas	[133]
miR-135a-5pmiR-124-3p	3′UTR	Glioblastoma	[134]

#### 6.1.1. Breast Cancer

In breast cancer, syntenin-1 is overexpressed [122,135,136]. Syntenin-1 interacts with TGF-β1 via its PDZ1 domain, which activates small GTPases such as RhoA and Cdc42, leading to EMT in breast cancer [82]. The expression of syntenin-1 in breast cancer is known to be controlled by two types of miRNAs. miR-135-5p inhibits the expression of syntenin-1 by directly binding to *SDCBP*, but its expression decreases in metastatic breast cancer, increasing migration and invasion by upregulating syntenin-1 expression [127]. miR-216b also inhibits syntenin-1 expression through direct interaction, and the migration and invasion abilities of breast cancer cells are reduced when miR-216b is overexpressed. The negative correlation between the expression levels of syntenin-1 and miR-216b was confirmed through patient sample analysis [128].

#### 6.1.2. Pancreatic Cancer

The expression of miR-216b is downregulated in pancreatic cancer compared to normal cells, leading to increased tumor size and tumor node metastasis progression [137]. miR-216b is also known to reduce the expression of KRAS, thereby inhibiting the progression of pancreatic cancer and inducing apoptosis [138]. In addition, miR-216b directly targets *SDCBP* to inhibit its expression, thereby regulating the EMT of pancreatic cancer through the inhibition of the PI3K pathway [129].

#### 6.1.3. Lung Cancer

Syntenin-1 expression is upregulated in lung cancer, and its overexpression is associated with poor prognosis in lung cancer patients [91,139]. miR-1273a enhances cisplatin sensitivity by directly targeting *SDCBP* to inhibit syntenin expression in lung cancer [130].

#### 6.1.4. Gastric Cancer

Syntenin-1 expression is also increased in gastric cancer compared to normal cells [122]. miR-361-5p is a cancer-suppressing miRNA that is inhibited during cancer development [140,141,142]. It inhibits syntenin-1 expression by binding with *SDCBP*, thereby promoting cancer growth and inhibiting apoptosis in gastric cancer. Additionally, a negative correlation between syntenin-1 and miR-361-5p was confirmed in gastric cancer patients [131].

#### 6.1.5. Melanoma

Syntenin-1 overexpression in melanoma cells induces a metastatic phenotype [55], while miR-23a is downregulated in metastatic melanoma cells, and its overexpression reduces migration and invasion ability [68,143]. Enhanced syntenin-1 levels in melanoma cells promote metastasis via c-Src activity [144]. miR-23a reduces syntenin-1 expression by binding to *SDCBP*, thereby inhibiting the progression and migration of melanoma cells through MAPK/ERK pathway inhibition [132].

#### 6.1.6. Gliomas and Glioblastoma

In contrast to syntenin-1, which is upregulated in glioma cells, miR-139-3p expression is reduced, and its reduction inhibits the growth, migration, and invasion of cancer cells by targeting syntenin-1 [133]. In glioblastoma, miR-135a-5p and miR-124-3p interact with *SDCBP* to inhibit syntenin-1 expression, thereby obstructing growth, migration, and infiltration [134].

### 6.2. The Regulation of miRNA by Syntenin

Syntenin-1 plays a crucial role in exosome formation and cargo sorting through syndecan and ALIX binding [34,111]. Exosomal microRNA, transmitted via exosomes, regulates essential processes such as cancer initiation, development, and metastasis in the tumor microenvironment [145]. One such cargo, onco-miRNA miR-494-3p, found in various cancers, targets the PI3K/AKT signaling pathway to promote lung cancer progression and regulates CXCR5 to promote cancer growth and EMT in glioblastoma, prostate cancer, and hepatocellular carcinoma [146,147,148,149,150]. Syntenin-1 expression in lung cancer cells increases the amount of exosome released, altering the delivery of miR-494-3p, a critical factor in regulating angiogenesis, tumor growth, and metastasis [23]. These studies suggest that syntenin-1-mediated exosome control plays a crucial role in cancer progression.

## 7. Conclusions

Syntenin, a scaffold protein containing two PDZ domains, plays a crucial role in various intracellular functions, depending on the binding partner. This review highlights the essential signaling pathways and functions of syntenin-1, with a specific focus on its involvement in exosome-mediated microRNA regulation. Exosomes are responsible for releasing cargoes such as microRNA, DNA, and proteins outside of the cell and delivering them to nearby cells. Syntenin-1 is involved in the formation and release of exosomes, which can cause changes in the surrounding cells. Recently, cell-to-cell communication through exosomes in the tumor microenvironment has been recognized as an important factor in the growth and EMT of cancer cells and is attracting attention as a new target for cancer treatment. Syntenin-1 is overexpressed in various tumors and acts as a crucial factor in cancer growth and metastasis. In addition to its direct impact on cancer development, syntenin-1 also induces mechanisms of metastasis and angiogenesis by altering other cells in the cancer microenvironment through exosomes. The relationship between syntenin-1 and microRNA, which act as oncogenes or tumor suppressors, offers potential new targets for cancer treatment and vesicle-based therapies. This review provides an overall understanding of syntenin-1’s function and summarizes recent studies on its role in cancer cells, suggesting various directions for future research.

## Figures and Tables

**Figure 1 ijms-24-09418-f001:**
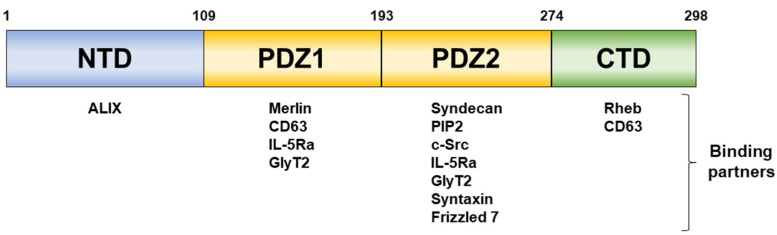
The four domains that compose syntenin and the proteins that bind to each domain. Each domain consists of the size up to the amino acid (aa) indicated (1–109 aa for NTD, 110–193 aa for PDZ1, 194–274 aa for PDZ2 and 275–298 aa for CTD).

**Figure 2 ijms-24-09418-f002:**
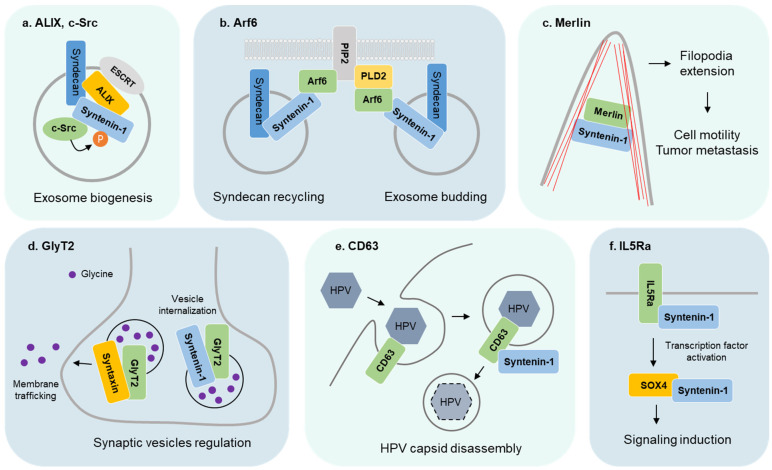
The binding partners of syntenin-1. (**a**) The combination of ALIX, c-Src, and syntenin-1 promotes the biogenesis and release of exosomes. (**b**) Arf6, a GTPase, interacts with syntenin-1 to help syndecan membrane trafficking or promotes exosome budding through the formation of PLD2-PIP2. (**c**) Merlin combines with syntenin-1 to move to the filopodia extension region and increases cell mobility and metastasis in cancer cells. (**d**) GlyT2 transports synaptic vesicles to the cell membrane through binding to syntaxin and helps release glycine. However, binding of syntenin-1 inhibits the combination of syntaxin and GlyT2, and the vesicles are internalized. (**e**) CD63 combines with HPV’s capsid protein to mediate endocytosis and promote trafficking and capsid disassembly through complex formation with syntenin-1. (**f**) The binding between syntenin-1 and IL5Ra promotes a signaling cascade and activates SOX4, a transcription factor. SOX4 avoids the degradation process and promotes gene transcription through direct binding to syntenin-1.

**Figure 3 ijms-24-09418-f003:**
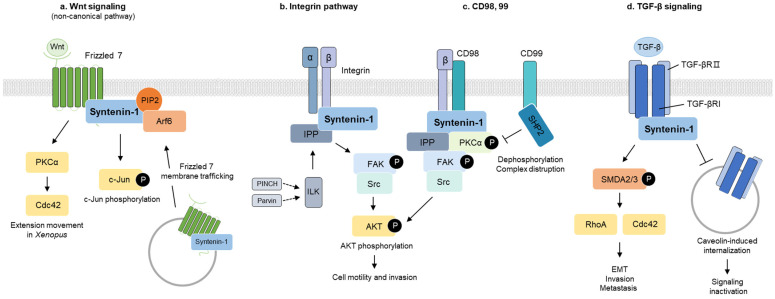
The signaling pathways regulated by combination with syntenin-1. (**a**) The non-canonical pathway of Wnt signaling is mediated by syntenin-1. The combination of syntenin-1 and the Frizzled 7 receptor increases membrane trafficking and promotes the activity of c-Jun and Cdc42. (**b**) The binding of the integrin receptor to syntenin-1 mediates the formation of the IPP complex and promotes cell motility and invasion through AKT activation. (**c**) CD98 activates the PKCα-FAK-Src-AKT signaling through binding with syntenin-1. CD99 inhibits syntenin-1-PKCα signaling via SHP2. (**d**) The binding of TGF-βR and syntenin-1 increases the activity of smad2/3-GTPase and inhibits the internalization of TGF-βR, which induces the metastatic ability of cancer cells.

**Figure 4 ijms-24-09418-f004:**
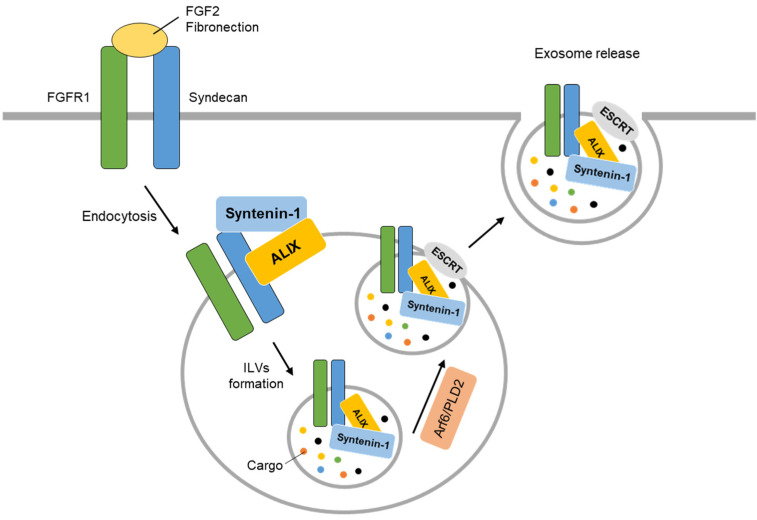
Syntenin-1 regulates the production and secretion of exosomes through interactions with ALIX, ESCRT, and GTPase. First, syntenin-1 forms ILVs with cargo through binding with ALIX and synthecan. It moves to the cell membrane through Arf6/PLD2, and buds are cut by ESCRT, which releases it into the extracellular space in the form of exosomes.

## Data Availability

Not applicable.

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
