# Peer review of "The Multifunctional Protein Syntenin-1: Regulator of Exosome Biogenesis, Cellular Function, and Tumor Progression"

_ijms, 2023, doi:10.3390/ijms24119418_

Round 1

Reviewer 1 Report

This review emphasized the role of syntenin-1 in exosome biogenesis and its interaction with exosomal biomarkers as well as exosomal cargos.  The authors listed a panel of binding partners of syntenin-1 which mediate the formation and release of exosomes as well as cargo loading onto exosomes. Different cellular signaling pathways were also illustrated. Overall, the review is informative. However, there are still some comments needed to be addressed.

1. Please check the manuscript and correct all typos. For example, Lin 59-63. The two sentences are exactly the same but with different references. interferon-y should be changed to interferon-gamma (Line 85). NTC should be NTD (Line 94)

2. The authors can give a table of the binding partners of syntenin-1, giving the name of the binding partner, the binding sites and signaling mechanisms. 

3. Line 253. The subtitle "4.3 CD98,99" doesn't seem to be aligned with the subtitles above. The authors can consider changing the subtitles.

4. The content of section 5.4.1 (Exosomes synthesis) largely overlaps with the introduction and section. 

Author Response

  1. Please check the manuscript and correct all typos. For example, Lin 59-63. The two sentences are exactly the same but with different references. interferon-y should be changed to interferon-gamma (Line 85). NTC should be NTD (Line 94)

We thank you for your helpful comments. We revised the sentences and words you pointed out and marked them as highlights. Also, we checked the typos overall.

  1. The authors can give a table of the binding partners of syntenin-1, giving the name of the binding partner, the binding sites and signaling mechanisms. 

Thank you for your precious comment for our manuscript. The inserted figure 2 and 3 summarize the binding partners and signaling pathway respectively. Therefore, we decided that the table you suggested would conflict with the existing figure summary.

  1. Line 253. The subtitle "4.3 CD98,99" doesn't seem to be aligned with the subtitles above. The authors can consider changing the subtitles.

We are grateful for your comment. As suggested by the reviewer, we revised the subtitle of 4.3 and can be found on line 252.

  1. The content of section 5.4.1 (Exosomes synthesis) largely overlaps with the introduction and section. 

We appreciate your constructive opinion. We intended to briefly mention that in the introduction part and to elaborate further on the overall process, including binding partners and mechanisms, in section 5.4.1. We modified the overlapping part in the introduction. It is expected that this modification will help the flow of the manuscript.

Reviewer 2 Report

This study is interesting with clinical significance. Exosomes are an important factor in intercellular communication and play an important in tumor metastasis . The authors made a comprehensive summary of the role of Syntenin-1. This is an excellent review with a strong focus and comprehensive scientific evidences. The followings are some comments to the authors.

Comments:

1. Please define all abbreviations in the text when used for the first time. For example, in line 14  PDZ.

2. I suggest listing a table to illustrate 6.2. The regulation of miRNA by Syntenin, such as Table 1.

3. Is there a error marker in Table 1 ( [127], Error! Bookmark not defined.[128])? Please confirm that.

Author Response

  1. Please define all abbreviations in the text when used for the first time. For example, in line 14 “ PDZ”.

We appreciate your helpful comment. We checked and corrected the abbreviation as a whole including PDZ.

  1. I suggest listing a table to illustrate “6.2. The regulation of miRNA by Syntenin”, such as Table 1.

We are grateful for your comment. Only one type of miRNA mentioned in section 6.2 is known, so we decided that it was inappropriate to add it as a table.

  1. Is there a error marker in Table 1 ( [127], Error! Bookmark not defined.[128])? Please confirm that.

We greatly thank the reviewer's meaningful comment. In the case of miR-216b, it is reported to have similar effects in both breast and pancreatic cancer. Table 1 has been modified to clearly confirm that [127] is a reference to breast cancer and [128] is a reference to pancreatic cancer.

Reviewer 3 Report

The review gives an overview of the synthenin-1.

The review describes in detail the syntenin structure and functions. There are many links with exosomes and how synthenin is implicated in release and signal cascade processes of the exosomes. I have not found a clear description on the involvement of synthenin in tumor invasion. Describe it better or propose another title.

The paragraphs in which the signal cascades are described, perhaps need to be schematized to be easier to understand.

This reference: Kugeratski FG, Hodge K, Lilla S, McAndrews KM, Zhou X, Hwang RF, Zanivan S, Kalluri R. Quantitative proteomics identifies the core proteome of exosomes with syntenin-1 as the highest abundant protein and a putative universal biomarker. Nat Cell Biol. 2021 Jun;23(6):631-641. doi: 10.1038/s41556-021-00693-y…………………….should be inserted in my opinion.

Author Response

  1. I have not found a clear description on the involvement of synthenin in tumor invasion. Describe it better or propose another title.

Thank you for your meaningful opinion. This manuscript shed light on various cellular function regulations, including exosome regulation by syntenin. It also summarized the function of syntenin as a regulator of cancer progression through signaling pathway and miRNA. According to the reviewer's opinion, the title has been revised to reflect the above contents.

  1. The paragraphs in which the signal cascades are described, perhaps need to be schematized to be easier to understand.

We appreciate your comment for our manuscript. We are schematically showing the contents of signaling cascades in figure 3. We think the explanation of signaling transmission is sufficiently supplemented through figure.

  1. This reference: Kugeratski FG, Hodge K, Lilla S, McAndrews KM, Zhou X, Hwang RF, Zanivan S, Kalluri R.Quantitative proteomics identifies the core proteome of exosomes with syntenin-1 as the highest abundant protein and a putative universal biomarker. Nat Cell Biol. 2021 Jun;23(6):631-641. doi: 10.1038/s41556-021-00693-y…………………….should be inserted in my opinion.

Thank you for suggesting an appropriate reference for our review. The suggested reference is that the amount of syntenin in exosomes isolated from various cells was the highest, and I think it is very meaningful to be included in our manuscript. We added the related contents to the session of '5.4.1 exosome synthesis'.

Reviewer 4 Report

This review summarises what is known about the protein syntenin-1 in terms of exosomes and cancer. This is generally fine, but there are some suggestions which would improve the quality of the manuscript.

- oncoproteins could be more clearly defined (line 35)

- it is stated that oncosomes are the largest EVs 1-10um in size (line 34), then later that apoptotic bodies are 50nm-200um (line 36). Need to correct this discrepancy.

- lines 59 - 63 contain repeating passages, and don't ready very clearly also.

- I don't think PDZ is defined, even though it is referred to a number of times?

- line 180, tetraspanin isn't one protein?

- line 238 should be integrin family

- there is some repetition, eg that syntenin-1 binds to syndecan, try to remove repetitions as there are a few.

- It is worth expanding on the role syndecans play in exosome release given the reviews perspective on exosomes.

- line 351 autophagy as a "self destructive process" does not read properly. Autophagy is a process of self degradation?

- line 418 - SDCBP is THE coding gene?

- table 1 correct the error

- structurally the review could be tidied. There are some sections that arn't really very informative - sometimes only one or two sentences, and reads more like a list. Some sections could be merged and the discussion made more engaging by defining what the relevance is of the points being made.

- line 478 what do you mean by scaffold protein?

Needs a proof-read to remove typographical and grammatical mistakes.

Author Response

  1. oncoproteins could be more clearly defined (line 35)

Thank you for the reviewer's comments. We revised it to oncogene, which is a clearer word, and you can check it with the highlight part of line 35.

  1. it is stated that oncosomes are the largest EVs 1-10um in size (line 34), then later that apoptotic bodies are 50nm-200um (line 36). Need to correct this discrepancy.

Thank you for your accurate point about the error. We checked the reference and revised it to 2um, and you can find the part to be modified in line 36.

  1. lines 59 - 63 contain repeating passages, and don't ready very clearly also.

We grateful for your constructive opinion for our manuscript. We have revised the paragraph you pointed out to remove repetitive content and make it more clear to understand.

  1. I don't think PDZ is defined, even though it is referred to a number of times?

We thankful for your appropriate point. We added the full name of PDZ in abstract section.

  1. line 180, tetraspanin isn't one protein?

Tetraspanin is a protein with four transmembrane domains and consists of more than 30 various families. We modified sentence of line 179 to express it more clearly.

  1. line 238 should be integrin family

There is no mention of the integrin family in line 238.

  1. there is some repetition, eg that syntenin-1 binds to syndecan, try to remove repetitions as there are a few.

Thank you very much for your meaningful comments. We revised the repetitive contents of syndecan in line 148 and protein function section.

  1. It is worth expanding on the role syndecans play in exosome release given the reviews perspective on exosomes.

We are grateful for your comment. This manuscript deals with the exosome biogenesis importantly, but emphasizes the importance of the role of syntenin in the process. Syndecan is also a key regulator of exosome production, but it was described focusing on syntenin to emphasize the syntenin function. We hope this answer is sufficient explanation for not extending the description about the syndecan.

  1. line 351 autophagy as a "self destructive process" does not read properly. Autophagy is a process of self degradation?

Thank you for your comment. Autophagy is sometimes described as self-eating or self-degradation, in the sense that cells remove unnecessary, misfunctional substances under stress. We revised the sentence of line 349 to help clear understanding.

  1. line 418 - SDCBP is THE coding gene?

We thankful for your comment. SDCBP (syndecan binding protein) is widely referred to as the coding gene name for syntenin.

  1. table 1 correct the error

Thank you for informing me of the error. We modified the miR-216b part because it is not accurately indicated that it affects two kinds of carcinoma.

  1. structurally the review could be tidied. There are some sections that arn't really very informative - sometimes only one or two sentences, and reads more like a list. Some sections could be merged and the discussion made more engaging by defining what the relevance is of the points being made.

Thank you for providing your valuable opinion for our manuscript. We revised the contents of ALIX, c-Src, and Arf6 proteins related to the production and secretion of exosomes in the 'binding partner' section, referring to the reviewer's comments. We have merged the relevant content and added some discussions. We hope the revised content fully reflects the reviewer's opinion.

  1. line 478 what do you mean by scaffold protein?

The scaffold protein is a protein that interacts with various proteins to help the signaling pathway. Syntenin also interacts with various proteins through the PDZ domain, so it is classified as a scaffold protein, so it was used as a word to refer this.